# QTSUMM: Query-Focused Summarization over Tabular Data

**Yilun Zhao[1]   Zhenting Qi[2]   Linyong Nan[1]   Boyu Mi[3]   Yixin Liu[1]**
**Weijin Zou[1]   Simeng Han[1]   Ruizhe Chen[3]   Xiangru Tang[1]   Yumo Xu[4]**
**Dragomir Radev[1]   Arman Cohan[1,5]**

[1]Yale University   [2]Harvard University   [3]Zhejiang University
[4]School of Informatics, University of Edinburgh   [5]Allen Institute for AI
`yilun.zhao@yale.edu`

## Abstract

People primarily consult tables to conduct data analysis or answer specific questions. Text generation systems that can provide accurate table summaries tailored to users' information needs can facilitate more efficient access to relevant data insights. Motivated by this, we define a new *query-focused table summarization* task, where text generation models have to perform human-like reasoning and analysis over the given table to generate a tailored summary. We introduce a new benchmark named QTSUMM for this task, which contains 7,111 human-annotated query-summary pairs over 2,934 tables covering diverse topics. We investigate a set of strong baselines on QTSUMM, including text generation, table-to-text generation, and large language models. Experimental results and manual analysis reveal that the new task presents significant challenges in table-to-text generation for future research. Moreover, we propose a new approach named REFAC-TOR, to retrieve and reason over query-relevant information from tabular data to generate several natural language facts. Experimental results demonstrate that REFACTOR can bring improvements to baselines by concatenating the generated facts to the model input. Our data and code are publicly available at `https://github.com/yale-nlp/QTSumm`.

## 1   Introduction

In the era of data-driven decision-making, tabular data plays a crucial role in facilitating data analysis, serving as a concise and structured representation of information (Kukich, 1983; Pasupat and Liang, 2015; Chen et al., 2020c; Zhu et al., 2021; Zhao et al., 2022a; Tang et al., 2023). People often consult tables to extract valuable insights and make informed decisions. For example, sales managers typically explore large tables with specific business questions to gain insights about clients and processes. Sports coaches will analyze performance

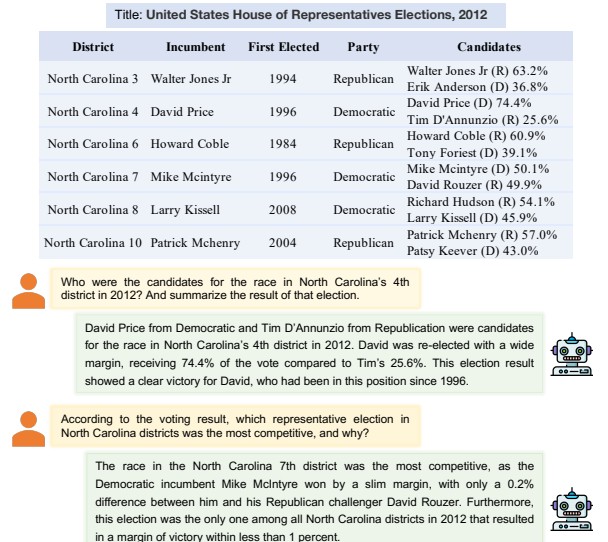

Figure 1: An example of QTSUMM. Given the numerous data points in the table, different users may be interested in various aspects for their own information-seeking or decision-making purposes. The system needs to perform human-like reasoning and analysis over relevant table regions to generate a tailored table summary.

tables containing various statistics to develop game strategies and make team adjustments. However, effectively accessing and comprehending the information contained within a large and complex table can be time-consuming for users (Hurst, 2000; Pasupat and Liang, 2015; Pujara et al., 2021; Nan et al., 2022a). Text generation systems that can accurately summarize a provided table according to users' information needs have the potential to greatly enhance data analysis and expedite the process of obtaining data insights.

Existing work and datasets on table-to-text generation (Parikh et al., 2020; Chen et al., 2020a; Cheng et al., 2022b; Lebret et al., 2016; Moosavi et al., 2021; Suadaa et al., 2021) have mainly focused on converting tabular data into coherent statements, aiming to present the structured data in a human-readable format. However, these approaches have overlooked the fundamental goal of addressing

users' *information-seeking* purposes. Table-to-text generation systems should adopt a more flexible and interactive approach that allows people to obtain a user-customized summary tailored to their information needs (Dang, 2006; Xu and Lapata, 2020; Zhong et al., 2021; Xu and Lapata, 2022; Zhou et al., 2023), as illustrated in Figure 1. While table question answering (QA) (Pasupat and Liang, 2015; Iyyer et al., 2017; Zhong et al., 2018; Chen et al., 2020c; Nan et al., 2022b) has made significant progress in answering fact-based questions, the primary focus of their approaches is on extracting relevant facts or entities from the table and composing short-form answers. Nevertheless, in real-world scenarios, users often have more complex and diverse information needs that extend beyond simple fact retrieval. They expect models to perform *human-like reasoning* and provide trustworthy explanations or analyses that accompany the extracted insights.

With comprehensive consideration of the real-world information needs of users when consulting tabular data, we propose a new task, *query-focused table summarization*. In this task, the model is required to generate a user-customized summary given the table and user query. To enable research in this area, we construct a human-annotated table-to-text generation dataset named QTSUMM[1], that contains 7,111 query-summary pairs over 2,934 Wikipedia tables covering diverse topics. Table 1 compares QTSUMM with previous table-to-text generation datasets. To the best of our knowledge, QTSUMM is the first dataset that tackles tasks of generating user-customized table summaries based on real-world scenarios.

We provide a comprehensive evaluation of current state-of-the-art models, including text generation (Lewis et al., 2020; Raffel et al., 2020; Chung et al., 2022), table-to-text generation (Liu et al., 2022b; Zhao et al., 2022b; Jiang et al., 2022), and large language models (Touvron et al., 2023a,b; Zheng et al., 2023; Jiang et al., 2023a; Xu et al., 2023; OpenAI, 2023). Our results and analysis from different perspectives reveal that the existing models struggle in solving this new task, highlighting the challenges the models face when performing human-like reasoning and analysis to generate summary tailored to users' information needs.

---

[1]We released the dataset at https://huggingface.co/datasets/yale-nlp/QTSumm using "gated repositories" to protect the data from automatic crawling (Jacovi et al., 2023).

To improve both text generation systems for QTSUMM, we propose REFACTOR. Given a user query, REFACTOR can retrieve and reason over query-relevant facts from the source table to generate multiple data insights in natural language sentences. Our results illustrate that directly concatenating the original input sequence with REFACTOR's generation can bring effective improvements to state-of-the-art baseline systems.

We conclude our main contributions as follows:

- We propose a new *query-focused table summarization* task, and construct a large-scale benchmark, QTSUMM, comprising 7,111 query-summary pairs collected in real-world situations. Strict quality control measures are employed to ascertain the high quality of the dataset.

- We conduct a systematic study of state-of-the-art models on QTSUMM, and illustrate that they are still far behind expert performance, motivating future research on this new table-to-text task.

- We present REFACTOR for the efficient retrieval and reasoning of query-relevant facts from tables. It demonstrates significant enhancements pertaining to state-of-the-art text generation baselines.

## 2 Related Work

**Table-to-Text Generation** As illustrated in Table 1, existing work and datasets on table-to-text generation typically pose the problem as either a single-sentence generation task (Chen et al., 2020a; Parikh et al., 2020; Cheng et al., 2022b; Liu et al., 2022a), or a generic summarization task (Lebret et al., 2016; Moosavi et al., 2021; Suadaa et al., 2021). In the *single-sentence generation* task (Parikh et al., 2020; Chen et al., 2020a; Cheng et al., 2022b), the focus is on generating fluent and faithful descriptions using provided table regions as a control for text generation. Nevertheless, using table regions for controlling text generation does not align with real-world scenarios, where people refer to tabular data for information-seeking purposes. The *generic table summarization* tasks (Lebret et al., 2016; Moosavi et al., 2021; Suadaa et al., 2021) aim to create concise and informative summaries based on the content of a given domain-specific table (i.e., sports or scientific). In contrast, the tables in QTSUMM cover diverse topics. Furthermore, considering the numerous data points in the table, various users may be interested in different aspects for their own information-seeking

| Dataset | Table Source | # Tables / Statements | # Words / Statement | Explicit Control | Rich in Analysis & Reasoning? |
|---|---|---|---|---|---|
| *Single-sentence Table-to-Text* | | | | | |
| ToTTo (Parikh et al., 2020) | Wikipedia | 83,141 / 83,141 | 17.4 | Table region | ✗ |
| LogicNLG (Chen et al., 2020a) | Wikipedia | 7,392 / 36,960 | 14.2 | Table regions | ✓ |
| HiTab (Cheng et al., 2022b) | Statistics web | 3,597 / 10,672 | 16.4 | Table regions & reasoning operator | ✓ |
| *Generic Table Summarization* | | | | | |
| RotoWire (Lebret et al., 2016) | NBA games | 4,953 / 4,953 | 337.1 | ✗ | ✗ |
| SciGen (Moosavi et al., 2021) | Sci-Paper | 1,338 / 1,338 | 116.0 | ✗ | ✓ |
| NumericNLG (Suadaa et al., 2021) | Sci-Paper | 1,355 / 1,355 | 94.2 | ✗ | ✓ |
| *Table Question Answering* | | | | | |
| FeTaQA (Nan et al., 2022b) | Wikipedia | 10,330 / 10,330 | 18.9 | Queries rewritten from ToTTo | ✗ |
| *Query-Focused Table Summarization* | | | | | |
| QTSumm | Wikipedia | 2,934 / 7,111 | 68.0 | Queries from real-world scenarios | ✓ |

Table 1: Comparison between QTSumm and existing table-to-text generation datasets.

purposes, making it challenging to create a generic summary that encompasses all the salient information within the table. Therefore, in this paper, we propose and investigate a new task setting related to *query-focused summarization*. FeTaQA (Nan et al., 2022b) is a table QA dataset that collects queries by rewriting ToTTo's (Parikh et al., 2020) statements into questions and uses the same statements as the answers. In comparison with FeTaQA, the queries in QTSumm were annotated under *real-world scenarios*, making them more natural and better-reflecting users' actual information needs.

**Reasoning Over Tabular Data** Enhancing the table reasoning capabilities of models is essential for a variety of tasks related to tables, such as table question answering (Pasupat and Liang, 2015; Iyyer et al., 2017; Zhong et al., 2018; Zhao et al., 2023d), table fact verification (Chen et al., 2020b), and table-to-text generation (Chen et al., 2020a; Cheng et al., 2022b). One prevalent approach is pre-training models with table-text joint reasoning data (Herzig et al., 2020; Liu et al., 2022b; Zhao et al., 2022b; Liu et al., 2022a; Jiang et al., 2022; Dong et al., 2022; Cheng et al., 2022a; Xie et al., 2022). Nevertheless, these models generate text in an end-to-end manner, resulting in reduced explainability and difficulties in handling more complex reasoning, such as arithmetic calculation. Therefore, we propose REFACTOR, which can retrieve and generate query-relevant facts from tables as intermediate results for model input (Zhou et al., 2022; Zhao et al., 2023b), mitigating the *implicit* reasoning processes of text generation models.

**Query-Focused Summarization** Initially formulated as a document summarization task, QFS aims to generate summaries from documents that are tailored to specific user queries (Dang, 2006). Despite its potential real-world applications, QFS remains a challenging task due to the lack of large-scale training data. Existing works have attempted to address this issue by leveraging distant NLP resources, including question answering (Xu and Lapata, 2020) and paraphrase identification (Su et al., 2020), and generic summarization (Xu and Lapata, 2022; Zhou et al., 2023). Recently, Zhong et al. (2021) adopted QFS for meeting summarization and proposed a human-annotated benchmark over meeting transcripts. Similar to text, effectively accessing and comprehending the information contained within a large and complex table can be time-consuming for users, while QFS remains unexplored in table-to-text generation. In this work, we extend QFS to this new modality for more effective information-seeking and decision-making purposes.

## 3 Query-Focused Table Summarization

### 3.1 Problem Formulation

We formally define the proposed query-focused table summarization task as follows. The input is a user query $\boldsymbol{Q}$, and a table $\boldsymbol{T}$. The table $\boldsymbol{T} = W \cup \{T_{i,j} | i \leq R_T, j \leq C_T\}$ has $R_T$ rows and $C_T$ columns, with $W$ being the table title and $T_{i,j}$ being the textual content in the $(i, j)$-th cell. The task objective of QTSumm is to generate a paragraph-long textual summary $\boldsymbol{Y} = (y_1, y_2, \dots, y_n)$ given the user query $\boldsymbol{Q}$ and source table $\boldsymbol{T}$:

$$\boldsymbol{Y} = \arg\max \prod_{i=1}^{n} P(y_i | y_{<i}, \boldsymbol{Q}, \boldsymbol{T}; \theta), \quad (1)$$

where $\theta$ denotes the parameters of a neural text generation model, and $y_i$ denotes the $i$-th tokens in the generated summary.

## 3.2 Data Collection Principles

At a high level, the goal of the data collection process is to obtain high-quality user queries and corresponding paragraph-long summaries grounded on the tabular data. We outline our key criteria for designing a benchmark to thoroughly evaluate the table-to-text summarization capabilities of models. To achieve this, we first design three principles for annotating a good query-summary pair:

- **Comprehensiveness**: The tailored summary should provide enough details and analysis of the source table to respond to the user query, fulfilling user's information need.

- **Attributablity & Faithfulness**: The query should be answerable using only information from the source table. The summary should be grounded on the source table, and not contain any unfaithful or nonsensical text.

- **Fluency**: Both the user query and its corresponding table summary should be coherent and fluent.

## 3.3 QTSUMM Annotation Pipeline

To ensure that QTSUMM annotation fulfills the aforementioned principles, we carefully design an annotation pipeline consisting of following steps:

**Source Table Collection**  QTSUMM uses tables from LOGICNLG (Chen et al., 2020a) and TOTTO (Parikh et al., 2020) datasets as source tables, as these tables are crwaled from Wikipedia and covers diverse domains and topics. We filter out tables that are 1) too large or too small, 2) with only string-type columns, or 3) with hierarchical structures (e.g., containing more than one table header). Then we randomly sample 2,000 candidate tables from LOGICNLG and TOTTO, respectively, for the query-summary annotation.

**User Query Annotation**  Given a table, the annotators are required to read its content, and determine whether the table is informative and intelligible to common web users. Then they were asked to come up with two or three queries, assuming they are users seeking certain information from the table. We require each query to be answerable using information only from the table. Moreover, as this work focuses on paragraph-long summaries

| Property | Value |
|---|---|
| Unique Tables | 2,934 |
| Query-Summary Pairs | 7,111 |
| Rows per Table (Median/Avg) | 10 / 11.8 |
| Columns per Table (Median/Avg) | 6 / 6.6 |
| Table Title Length (Median/Avg) | 7 / 7.6 |
| Query Length (Median/Avg) | 22 / 22.3 |
| Relevant Rows (Median/Avg) | 4 / 3.8 |
| Summary Length (Median/Avg) | 63 / 68.0 |
| Training Set Size (Table/Summary) | 2,055 / 4,981 (70%) |
| Development Set Size (Table/Summary) | 439 / 1,052 (15%) |
| Test Set Size (Table/Summary) | 440 / 1,078 (15%) |

Table 2: Basic statistics of QTSUMM dataset.

as query responses, we avoid queries that can be answered in a short sentence (e.g., "Which country held the 2022 FIFA World Cup?").

**Query-Focused Summary Annotation**  Given a table and user query, we ask another annotator to use only information from the source table to write a paragraph-long summary that satisfies the user's information need. We encourage annotators to produce sophisticated summaries that 1) contain as much information from the table as possible, and 2) involve more types of reasoning over multiple relevant table regions. To further encourage high quality annotations, we adopt the "two channel collection" design (Chen et al., 2020b), in which the annotators would be paid 60% more if their summaries are manually verified to exhibit adequate complexity. We also require the annotators to annotate the row indices of relevant table regions that are referenced in the written summary, allowing future researchers to quantify how well the summaries are grounded in the table in their work.

**Multi-Round Validation**  We conduct a multi-round validation protocol to ensure that the annotated data fulfills the aforementioned annotation principles. We first assign query annotators to validate each summary against their corresponding queries, and fix the mistakes if there are any. Then we check 1) whether a query-summary pair contain adequate information and complex aggregation by examining the length of the summary, and 2) whether the information in summary is essential in responding to the user query. We manually revise pairs that do not meet the above standard.

## 3.4 Annotation Quality Control

Table 2 describes the basic statistics of QTSUMM. In addition to the multi-round validation, we carefully design several quality control approaches,

comprising expert annotation and numerous annotation de-biasing designs, to ensure the high quality of QTSUMM annotations.

**Expert Annotators** To help improve the annotation process, five experts with professional experience in the text summarization tasks are invited to conduct the *internal annotation*. They are asked to provide feedback regarding the task instructions and the user experience of the annotation interface, based on which we iteratively modify the annotation guideline and interface design. In the stage of *external annotation*, we enroll 17 graduate students majoring in STEM fields (10 females, and 7 males). We do not use the crowd-source annotation platform such as Mechanical Turk as our preliminary study indicates that annotators on MTurk fail to annotate high-quality query-summary data. Before starting the official annotation process, each annotator is given a two-hour training session to learn the annotation requirements and interface.

**Annotation De-biasing** We observed several kinds of annotation bias during our internal annotation, and we proposed countermeasures as follows for annotation de-biasing:

*Source Table Diversity:* During internal annotation, we found that many tables in LOGICNLG have similar content. For example, there are around 200 tables describing the results of football games, with identical table headers. To ensure the diversity of source tables, we keep only one table for each unique table header.

*Query Diversity:* When annotating queries, annotators may prefer simpler ones, resulting in low query diversity. Therefore, we frequently monitor the diversity of queries for each annotator. Annotators are also encouraged to craft queries that are either creative or require complex reasoning in summarization, resulting in a doubled payment to compensate them for the extra time.

*Supporting Fact Position:* We found that annotators prefer to raise queries regarding the first few rows of each table. To deal with such bias regarding supporting fact positions, we randomly highlight certain rows for each table in the annotation interface. We require the annotators to write queries whose summaries should cover at least two rows of the highlighted regions.

We also report the human evaluation scores and inter-evaluator agreements over 200 sampled query-summary pairs. QTSUMM has a high annotation

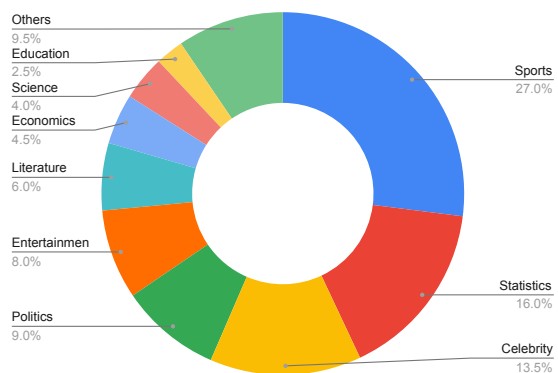

Figure 2: Domain distribution of QTSUMM tables.

| Annotation Quality | %S ≥ 4 | Agree | Kappa / 95% CI |
|---|---|---|---|
| Table Informativeness | 84.9 | 0.81 | 0.77 / [0.72, 0.82] |
| Query Meaningfulness | 93.2 | 0.89 | 0.84 / [0.79, 0.89] |
| Query Complexity | 91.4 | 0.87 | 0.81 / [0.75, 0.87] |
| Query Fluency | 97.2 | 0.94 | 0.92 / [0.90, 0.94] |
| Relevant Rows Correctness | 89.7 | 0.85 | 0.83 / [0.79, 0.88] |
| Summary Comprehensiveness | 97.5 | 0.97 | 0.93 / [0.90, 0.96] |
| Summary Faithfulness | 91.6 | 0.90 | 0.88 / [0.84, 0.92] |
| Summary Fluency | 96.1 | 0.93 | 0.89 / [0.86, 0.92] |

Table 3: Human evaluation over 200 samples of QT-SUMM. Three internal evaluators were asked to rate the samples on a scale of 1 to 5. We report 1) percent of samples that have an average score ≥ 4 to indicate the annotation quality of QTSUMM; and 2) percent of agreement and Randolph's Kappa with 95% CI (Randolph, 2005) to show the inter-annotator agreement.

quality and inter-annotator agreement (Table 3).

### 3.5 QTSUMM Evaluation

We develop a comprehensive approach for evaluating QTSumm, incorporating both automated and human evaluation. We adopt following popular automated evaluation metrics:

**BLEU** (Papineni et al., 2002) computes the geometric average of the precision over output text's n-grams. We used SacreBLEU (Post, 2018) that produces comparable and reproducible BLEU scores.

**ROUGE** (Lin and Hovy, 2003) measures the word overlap between the candidate and reference summaries. We reported F1 score for ROUGE-L (longest common subsequences).

**METEOR** (Banerjee and Lavie, 2005) is based on a generalized concept of unigram matching between the machine-produced translation and human-produced reference translations.

**BERTScore** (Zhang et al., 2020) computes the sim-

ilarity between the reference and generated summary using contextual word embeddings.

**TAPAS-Acc** (Herzig et al., 2020; Liu et al., 2022a) is a reference-free metric that uses TAPAS (Herzig et al., 2020) fine-tuned on the Tab-Fact dataset (Chen et al., 2020b) as the backbone to evaluate the faithfulness of generation.

**AutoACU** (Liu et al., 2023a) is an interpretable and reference-based summarization evaluation system that exhibits better alignment with human judgements. The A2CU first extracts atomic content units (ACUs) from the generated summary and then evaluates them against reference. A3CU is an accelerated version of A2CU that directly computes the similarity between two text without extracting ACUs, but with the similar evaluation target. We use F1 score of A3CU for evaluation.

For **human evaluation**, the summaries from different models were evaluated by experts from three criteria (i.e., *comprehensiveness*, *faithfulness*, and *fluency*) that have been discussed in Section 3.2. Each summary was scored from 1 (worst) to 5 (best) for each criteria, with the final score averaged across different evaluators.

## 4   REFACTOR

QTSUMM requires models to perform human-like reasoning in generating summaries that provide comprehensive and precise analysis of the source table to fulfill the user's information need. However, existing end-to-end text generation models rely on error-prone *implicit* reasoning processes for generating text, leading to diminished explainability and challenges in addressing user queries that necessitate complex human-like reasoning (Zhou et al., 2022; Zhao et al., 2023b). To address this, we present REFACTOR, to retrieve and reason over query-relevant information from tabular data to generate several NL data insights (i.e., facts) as *explicit* reasoning results. As shown in Figure 3, the generated facts is concatenated to the model input to mitigate the *implicit* reasoning issues, enhancing the comprehensiveness and faithfulness of generated summary. We next discuss the implementation of REFACTOR.

### 4.1   Fact Generation

Given the user query and source table, REFACTOR will generate several candidate facts by executing various forms of human-like reasoning over the ta-

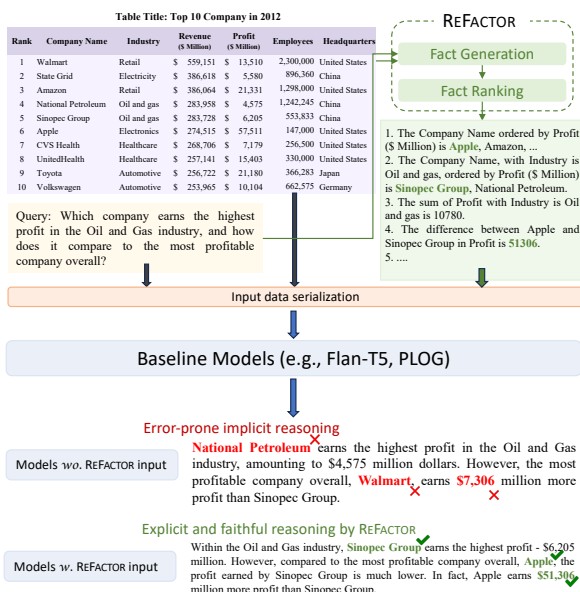

Figure 3: Enhancing fine-tuned models with the proposed REFACTOR. After generating and selecting the top-$n$ query-relevant facts obtained through various reasoning operations (e.g., numerical comparison, counting), these facts are concatenated with query and table data as the model input in both fine-tuning and inference stage. REFACTOR can mitigate the error-prone implicit reasoning issues of end-to-end text generation systems. For LLM in zero- or few-shot setting, we provide generated facts within the prompts (Figure 5 in Appendix A).

ble. Specifically, we define 6 types of table reasoning operations (e.g., numerical operation, counting, and conjunction) that are necessary for the QT-SUMM task, as shown in Table 7 in the Appendix. For each reasoning operation, the fact generator (adopted from Zhao et al. (2022b)) takes a table and a query as input. It produces multiple facts based on the fact template. Each fact template includes several placeholders that need to be filled with information retrieved from the table. Specifically, column `col` and cell value `val` are indexed to specify the column and cell name, respectively. Some templates also regulate that the selected column and cell value must be date or number type. `OPERATOR` corresponds to operators that are instantiated according to the specific reasoning reasoning. And `CONDITION:i` can be 1) a cell value from the `i`-th column; or 2) a number/temporal comparison statement if the `i`-th column is date or number type. After substituting all the placeholders in the provided template, the fact generator will programmatically return the `executed_results` and form one fact. Once facts for a {table, query} pair are collected from different fact generators, we pass them to the Fact Ranking process.

## 4.2 Fact Ranking

Given the query and source table, each fact generator will be utilized to generate several query-relevant facts, resulting in a large number of candidate facts in total. Therefore, we need to rank the generated facts to select the most relevant ones. We use the QA encoding model (Reimers and Gurevych, 2019) to obtain the embedding of the query and each generated fact. Then, we select the top-$n$ generated facts with the highest cosine similarity to the query embedding. In practice, we assign $n$ as $max(\sqrt{\frac{row_{num} \times column_{num}}{2}}, 5)$, and ensure that the number of selected facts from each type of reasoning operation does not exceed 3. The selected facts, which are handy and readily available for end-to-end text generation systems, are then concatenated into the model input.

## 5 QTSUMM Experiments

### 5.1 Baseline Systems

We evaluate the following three types of state-of-the-art baseline systems[2] on QTSUMM:

### 5.1.1 Text Generation Models

**BART** (Lewis et al., 2020) is a pre-trained denoising autoencoder with transformer-based architecture and shows effectiveness in NLG tasks.

**T5** (Raffel et al., 2020) demonstrates effectiveness in NLG tasks by treating all NL problems as text-to-text tasks during pre-training stage.

**Flan-T5** (Chung et al., 2022) enhances T5 by scaling instruction fine-tuning and demonstrates better human-like reasoning abilities than the T5.

### 5.1.2 Table-to-Text Generation Models

**TAPEX** (Liu et al., 2022b) continues pre-training the BART model by using a large-scale corpus of synthetic SQL query execution data. It shows better table understanding and reasoning abilities.

**ReasTAP** (Zhao et al., 2022b) enhances the table understanding and reasoning abilities of BART by pre-training on a synthetic Table QA corpus.

**OmniTab** (Jiang et al., 2022) uses the same backbone as TAPEX, and is further pre-trained on collected natural and synthetic Table QA examples.

### 5.1.3 Large Language Models

**Llama-2**[3] (Touvron et al., 2023a,b) is an open-source large language model trained on large-scale and publicly available datasets.

**Vicuna**[4] (Zheng et al., 2023) is tuned from Llama-1 with instruction-following data, exhibiting better instruction-following capabilities.

**Mistral**[5] (Jiang et al., 2023a) is a 7–billion-parameter LLM that outperforms Llama-2-13B across most of popular evaluated benchmarks.

**Lemur**[6] (Xu et al., 2023) is tuned from Llama-2 with instruction-following data, exhibiting competitive natural language and coding capabilities.

**GPT** (Brown et al., 2020; OpenAI, 2023) is a powerful large language model which is capable of generating human-like text and performing a wide range of NLP tasks in a few-shot setting.

### 5.2 Experimental Setup

The specifics of input data serialization and LLM prompting examples are discussed in Appendix A. All experiments were conducted on an 8 NVIDIA RTX A6000 48GB cluster. We selected the `large` version for all fine-tuned baseline models, whose weights are publicly available at HuggingFace. For each fine-tuning experiment, we ran 15 epochs with a batch size of 128. The best fine-tuning checkpoints were selected according to the validation loss. The experiments for open-sourced LLMs were conducted using `vLLM` framework (Kwon et al., 2023). We used `gpt-3.5-turbo-0613` for GPT-3.5 and `gpt-4-0613` for GPT-4 via the OpenAI APIs[7]. For LLM hyperparameter settings, we set temperature as 1.0, Top P as 1.0, and maximum output length as 256.

### 5.3 Main Results

We draw following conclusions based on the automated and human evaluation results (Table 4 & 6).

**Importance of table structure understanding** Table-to-text generation models achieve better performance than their corresponding text-generation

---

[2]We released the model weights of evaluated fine-tuned models at HuggingFace (https://huggingface.co/yale-nlp/{model_name}-finetuned-qtsumm).

[3]https://huggingface.co/meta-llama/llama-2-{size}b-chat-hf

[4]We only evaluate Vicuna (https://huggingface.co/lmsys/vicuna-33b-v1.3) under zero- and one-shot settings, as some examples under the two-shot setting might exceeds its maximum length limit.

[5]mistralai/Mistral-7B-Instruct-v0.1

[6]https://huggingface.co/OpenLemur/lemur-70b-chat-v1

[7]https://openai.com/api/

| Type | Model | Backbone | Avg Len | BLEU | ROUGE-L | METEOR | BERTScore | TAPAS-Acc | **A3CU** |
|---|---|---|---|---|---|---|---|---|---|
| Ground Truth | | | 67.8 | | | | | | |
| Text Generation *fine-tuning* | T5-large | – | 61.8 | 20.3 | 38.7 | 40.2 | 89.6 | 75.1 | 43.5 |
| | Flan-T5-large | T5 | 74.1 | 19.9 | 39.8 | 42.5 | 89.8 | **83.9** | 46.3 |
| | BART-large | – | 60.1 | 21.2 | 40.6 | 43.0 | **90.6** | 77.1 | 48.0 |
| | *w.* REFACTOR | – | 61.4 | **21.5** (+0.3) | **41.0** (+0.4) | **43.1** (+0.1) | 90.1 (-0.5) | 79.4 (+2.3) | **48.6** (+0.6) |
| Table-to-Text *fine-tuning* | ReasTAP | BART | 61.4 | 22.5 | 41.9 | 44.3 | 90.8 | 80.6 | 51.9 |
| | TAPEX | BART | 70.1 | **23.1** | 42.1 | **45.6** | 90.6 | **87.8** | 52.0 |
| | OmniTab | BART | 59.5 | 22.4 | **42.4** | 44.7 | **91.0** | 80.2 | 53.1 |
| | *w.* REFACTOR | BART | 58.3 | 22.5 (+0.1) | 42.2 (-0.2) | 44.8 (+0.1) | 90.7 (-0.3) | 80.3 (+0.1) | **54.0** (+0.9) |
| LLM *zero-shot* | Llama-2-13B | – | 64.3 | 14.6 | 25.5 | 30.9 | 86.8 | 76.6 | 28.6 |
| | Llama-2-7B | – | 110.3 | 13.3 | 31.3 | 42.5 | 88.8 | 78.1 | 37.3 |
| | Mistral-7B | Llama-2 | 98.4 | 13.8 | 31.7 | 41.4 | 89.1 | 73.0 | 37.5 |
| | *w.* REFACTOR | Llama-2 | 99.0 | 13.8 (+0.0) | 31.4 (-0.3) | 41.5 (+0.1) | 88.7 (-0.4) | 74.5 (+1.5) | 37.7 (+0.2) |
| | Vicuna-33b | Llama-1 | 93.8 | 15.1 | 32.6 | 42.2 | 89.2 | 82.0 | 40.0 |
| | Lemur-70B | Llama-2 | 102.3 | 13.3 | 30.9 | 39.9 | 87.8 | 82.8 | 40.8 |
| | Llama-2-70B | – | 91.5 | 17.2 | 35.2 | 44.1 | 89.8 | 85.7 | 45.7 |
| | *w.* REFACTOR | – | 98.1 | 17.0 (-0.2) | 34.7 (-0.5) | 44.6 (+0.5) | 90.0 (+0.2) | 82.3 (-3.4) | 46.3 (+0.6) |
| | GPT-3.5 | – | 82.5 | **21.1** | **40.7** | 49.1 | **91.1** | 89.7 | 55.5 |
| | *w.* REFACTOR | – | 85.4 | 20.6 (-0.5) | 39.8 (-0.9) | **49.2** (+0.1) | 90.5 (-0.6) | 89.9 (+0.2) | 55.9 (+0.4) |
| | GPT-4 | – | 86.9 | 19.8 | 38.4 | 48.4 | 85.8 | **92.3** | 57.5 |
| | *w.* REFACTOR | – | 88.2 | 19.6 (-0.2) | 37.9 (-0.5) | 48.1 (-0.3) | 87.1 (+1.3) | **92.3** (+0.0) | **57.5** (+0.0) |
| LLM *1-shot* | Llama-2-13B | – | 61.5 | 13.8 | 23.9 | 28.1 | 86.6 | 81.4 | 26.5 |
| | Mistral-7B | Llama-2 | 96.6 | 13.7 | 31.5 | 40.9 | 88.9 | 71.7 | 36.7 |
| | *w.* REFACTOR | Llama-2 | 94.2 | 14.1 (+0.4) | 31.8 (+0.3) | 40.7 (-0.2) | 88.9 (+0.0) | 72.2 (+0.5) | 38.2 (+1.5) |
| | Llama-2-7B | – | 105.0 | 13.6 | 32.3 | 42.5 | 89.1 | 75.3 | 38.5 |
| | Lemur-70B | Llama-2 | 86.5 | 14.3 | 31.5 | 38.3 | 88.1 | 81.3 | 39.8 |
| | Vicuna-33b | Llama-1 | 75.0 | 19.3 | 37.0 | 43.8 | 90.1 | 78.4 | 45.3 |
| | Llama-2-70B | – | 92.8 | 18.2 | 37.3 | 46.2 | 90.2 | 86.5 | 48.1 |
| | *w.* REFACTOR | – | 92.0 | 18.1 (-0.1) | 37.0 (-0.3) | 46.2 (+0.0) | 90.3 (+0.1) | 86.7 (+0.2) | 48.3 (+0.2) |
| | GPT-3.5 | – | 88.0 | 20.2 | **40.0** | 49.7 | 90.9 | 91.7 | 55.6 |
| | *w.* REFACTOR | – | 85.2 | **20.3** (+0.1) | 39.8 (-0.2) | 50.0 (+0.3) | 91.2 (+0.3) | 92.2 (+0.5) | 57.0 (+1.4) |
| | GPT-4 | – | 92.1 | 19.0 | 39.9 | 51.2 | 91.0 | **94.3** | 60.1 |
| | *w.* REFACTOR | – | 89.4 | 19.5 (+0.5) | **40.0** (+0.1) | **51.4** (+0.2) | **91.3** (+0.3) | 93.7 (-0.6) | **61.3** (+1.2) |
| LLM *2-shot* | Llama-2-13B | – | 72.6 | 17.5 | 31.2 | 37.3 | 88.6 | 81.2 | 37.1 |
| | Mistral-7B | Llama-2 | 86.0 | 14.9 | 32.7 | 40.7 | 89.1 | 72.8 | 38.4 |
| | Llama-2-7B | – | 99.3 | 14.0 | 33.2 | 42.3 | 89.0 | 77.9 | 39.6 |
| | Lemur-70B | Llama-2 | 82.7 | 15.0 | 32.0 | 38.5 | 88.4 | 81.6 | 40.6 |
| | Llama-2-70B | Llama-2 | 87.3 | 19.0 | 38.0 | 46.4 | 90.4 | 87.3 | 49.1 |
| | GPT-3.5 | – | 89.8 | **20.0** | 39.9 | 50.0 | 90.9 | 93.2 | 56.2 |
| | GPT-4 | – | 90.1 | 19.5 | **40.5** | **51.1** | **91.1** | **93.3** | **61.0** |

Table 4: Automated evaluation results on the QTSUMM test set, involving three types of baseline systems with and without REFACTOR. We used `chat` or `instruct` version for each type of LLMs. Within each experimental setting, we used A3CU (F-score) as the ranking indicator of model performance. Due to the budget constraints, for all LLM *w.* REFACTOR experiments, we randomly selected 200 samples.

| # Examples | Error Types | Representative Question | Explanation |
|---|---|---|---|
| 24 / 200 | Difficulty in parsing cell values via rule-based methods | | The relevant numeric- or time-type columns are hard to parse (e.g., multiple numbers and text within one cell), thus REFACTOR fail to generate related facts. |
| 17 / 200 | Complex user query causes difficulty in ranking related facts | Analyze the correlation between the size of the geographical area of a Gmina type and its population? | REFACTOR employs the QA encoding model for fact ranking. However, it struggles to understand complex information needs from users, such as the "correlation between A and B", and might consequently rank irrelevant facts higher. |
| 13 / 200 | Unsupported reasoning operations | Who are the top three coaches with the highest win percentages? Analyze their performance in the 2019-2020 season. | The table only contains "wins" and "overall games" columns. Models must compute the winning percentages independently. However, REFACTOR does not support such rate calculations |
| 5 / 200 | Other errors | | |
| 141 / 200 | Successful cases | | |

Table 5: Case study on REFACTOR's failure cases.

| Model | Faithfulness | Compre. | Fluency |
|---|---|---|---|
| BART | 3.26 | 3.67 | 4.56 |
|   *w*. REFACTOR | 3.37 (+0.11) | 3.72 (+0.05) | 4.59 (+0.03) |
| OmniTab | 3.30 | 3.58 | 4.52 |
|   *w*. REFACTOR | 3.45 (+0.15) | 3.69 (+0.11) | 4.52 (+0.0) |
| 1-shot Mistral-7B | 2.98 | 3.77 | 4.65 |
|   *w*. REFACTOR | 3.12 (+0.14) | 3.82 (+0.05) | 4.52 (-0.13) |
| 1-shot Llama-2-70B | 3.08 | 3.82 | 4.69 |
|   *w*. REFACTOR | 3.36 (+0.28) | 3.99 (+0.17) | 4.66 (-0.03) |
| 0-shot GPT-3.5 | 3.65 | 3.94 | 4.66 |
|   *w*. REFACTOR | 3.84 (+0.19) | 4.03 (+0.09) | 4.74 (+0.08) |
| 0-shot GPT-4 | 3.92 | 4.12 | 4.84 |
|   *w*. REFACTOR | 4.08 (+0.16) | 4.15 (+0.03) | 4.70 (-0.14) |
| 1-shot GPT-3.5 | 3.84 | 4.20 | 4.86 |
|   *w*. REFACTOR | 3.95 (+0.11) | 4.27 (+0.07) | 4.84 (-0.02) |
| 1-shot GPT-4 | **4.11** | 4.32 | **4.88** |
|   *w*. REFACTOR | 4.08 (-0.03) | **4.35** (+0.03) | 4.76 (-0.12) |

Table 6: Human evaluation results (Likert Scale Scoring) of selected baselines on the test set. Five experts are enrolled to evaluate 50 predictions for each model.

backbones, demonstrating the importance of considering table structure for the QTSUMM task.

**Importance of reasoning and analysis** Among text generation models, Flan-T5, which enhances T5 through scaled instruction fine-tuning, outperforms T5. Moreover, LLMs with improved reasoning capabilities (i.e., Llama-2-70B and GPT-4) also achieve better performance. These findings indicate the significance of reasoning and analytical skills in handling the QTSUMM task.

**Mismatch between automated and human evaluation** Despite receiving low scores in popular automated evaluation metrics such as BLEU and ROUGE, GPT-* exhibit better performance than state-of-the-art fine-tuned models in human evaluation. This finding underscores the need for future research to investigate the development of automated evaluation metrics for the QTSUMM task that better align with human judgments (Zhang and Bansal, 2021; Liu et al., 2023a; Jiang et al., 2023b).

**Effectiveness of REFACTOR** As assessed by human evaluation, baseline systems employing REFACTOR typically yield better performance, especially in faithfulness-level. This suggests the efficacy of REFACTOR in enhancing the reasoning process in text generation.

### 5.4 Error Analysis

For a deeper understanding of the query-focused table summarization task on QTSUMM, we conduct an error analysis to illustrate existing challenges.

We identify four common mistakes that current text generation models are likely to make (i.e., **hallucination**, **factual incorrectness**, **user intent misunderstanding**, and **repetition**), providing detailed examples and explanations for each type of common mistake in Table 8 in the Appendix.

### 5.5 REFACTOR Analysis

We also undertake a human evaluation to examine the efficacy of REFACTOR in generating query-relevant facts from tabular data. Specifically, we randomly sample 200 examples from QTSUMM validation set, and ask two human evaluators to evaluate each fact generated by REFACTOR, determining its relevance to the query. 56.4% generated facts (528 out of 937) are labeled as "relevant", suggesting an adequate coverage of REFACTOR. To delve deeper into this, we also conduct a case study examining the failure cases, specifically those examples where less than two facts were annotated as "relevant". We identified three kinds of common failure cases: (1) difficulty in parsing cell values via rule-based methods, (2) complex user query causes difficulty in ranking related facts, and (3) unsupported reasoning operations. We provide detailed examples and explanations in Table 5.

## 6 Conclusion

This paper defines a new query-focused table summarization task, and constructs a large-scale benchmark, QTSUMM. We investigate a set of strong baselines, including text generation, table-to-text generation, and large language models. Experimental results and manual analysis reveal that the new task presents significant challenges in table-to-text generation. Moreover, we propose a novel approach named REFACTOR, to retrieve and reason over query-relevant information from tables, improving the faithfulness of generated summary.

## Acknowledgements

We would like to dedicate this paper to the memory of Dr. Dragomir Radev. Dr. Radev's leadership, guidance, and expertise were instrumental in shaping the direction and quality of this project. We appreciate the efforts of all annotators in constructing QTSUMM and conducting human evaluation. We are grateful to the Google TRC program for their support. We would also like to thank the anonymous reviewers and action editors for constructive discussions and feedback.

## Limitations and Future Work

The baseline systems provided have a restricted maximum number of tokens they can accommodate (e.g., 1024 for all examined fine-tuned models), which prevents them from generating summaries for large tables that, when converted into a sequence, exceed the maximum number of tokens. To handle large tables (e.g., with more than 300 table cells), future work can apply neural models (Herzig et al., 2020; Liu et al., 2022b) to first filter out those query-irrelevant rows or columns.

Moreover, this paper demonstrates the effectiveness of using intermediate results obtained from explicit reasoning operations to mitigate the implicit reasoning issues. However, the proposed REFACTOR utilizes template-based method to generate facts. Although such template-based approach can ensure the factual correctness of generated facts, as discussed in Section 5.5, it might not cover all crucial facts for some complex user query. We believe following directions warrant further exploration: (1) *Complex query decomposition*. Our case study reveals that the TAPEX-based fact ranking module struggles with comprehending complex questions. To address this, future research could investigate LLM chain-of-thought methods to break down complex questions into more understandable and actionable sub-questions. (2) *Tool usage*. The predefined and template-based execution modules in the REFACTOR fact generation phase have their limitations. Recent studies (Schick et al., 2023; Lu et al., 2023; Paranjape et al., 2023; Gou et al., 2023; Qiao et al., 2023) highlight the impressive abilities of LLMs in making and utilizing tools for problem-solving. It would be intriguing to explore if LLMs can produce executable programs from scratch to derive query-relevant insights. (3) *Explainable automated evaluation*. In Section 5.3, a discrepancy between automated and human evaluation results is observed. Such discrepancies are concerning, as developers might opt for suboptimal systems for real-world applications if they solely rely on automatic metrics for comparing and ranking different text generation systems. Therefore, a more reliable and explainable automated evaluation system is required (Zhang and Bansal, 2021; Liu et al., 2023a,b; Jiang et al., 2023b).

## Ethical Consideration

The source tables in QTSUMM were collected from LOGICNLG (Chen et al., 2020a) and TOTTO (Parikh et al., 2020) datasets, which are publicly available under the MIT license[8] and CC BY-SA 3.0 license[9], respectively. They both permit us to compose, modify, publish, and distribute additional annotations upon the original dataset.

For the external annotation of QTSUMM, we hired 17 graduate students majoring in STEM majors. We regard 1) creating three queries for one table, and validating the corresponding summaries annotated by others, and 2) composing a query-focused summary response as a unit task. And we paid around $1.5 for each unit task. For creative annotation rewards, we paid additional $0.5 for a query, and $1.5 for a summary. Averagely, an annotator can finish 7 unit tasks per hour after training and practicing. And the hourly rates are in the range of $9 and $13 based on the different working speed (above the local average wage of similar jobs). We recommended that annotators complete a maximum of 30 unit tasks per day in order to reduce pressure and maintain a comfortable pace. In total, the approximate working hours to annotate QTSUMM dataset was 1,400 hours. The whole annotation work lasted about 40 days.

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

## A Implementation Details

**Input Data Serialization** The input contains a user query, and corresponding table data. For text generation and large language models (Section 5.1.1 & 5.1.3), we followed recent works on table-to-text generation (Liu et al., 2022b; Xie et al., 2022; Zhao et al., 2023c,a) to flatten the table data as T=[HEADER]:h, [ROW]1:$r_1$,...,[ROW]n:$r_n$, where h is table header, $r_i$ is the i-th table row. For text generation models, [HEADER] and [ROW] are special tokens indicating the region of table headers and rows respectively; while for LLMs, we set them as empty strings. We also separated headers or cells in different columns using a vertical bar |. In this way, the flattened table input can be fed directly into text generation models. For table-to-text generation models (Section 5.1.2), we followed their original data processing methods to input the query and table data.

| Reasoning | Example of Fact Templates | Example of Fact |
|---|---|---|
| Conjunction | The `col` that have `CONDTION` are `executed_results`. | The **Player Name** that have **Country** is *Canada* are `Corey Conners, Nick Taylor, Adam Svensson`. |
| Counting | `executed_results col:1` have `col:2 CONDITION:2`. | 2 **Game** have **Attendance** *greater than 10,235*. |
| Temporal or Numerical Order | The `col:1` ordered by `col:3` are `executed_results`.
The `col:1`, with `col:2 CONDITION:2`, ordered by `col:3` are `executed_results`. | The **Company** ordered by **Sales** are `Apple, Nvidia, Google, (...abbreviate...)` |
| Temporal or Numerical Comparison | The `col:1` that `col:2 CONDITION:2` are `executed_results`. | The **institutions** that **Founded year** *is earlier than 1860* are `Adrian College, Michigan State University`. |
| Numerical Operation (`Sum, Avg`) | The `OPERATOR` of `col:1` with `col:2 CONDITION:2` is `executed_results`. | The *sum* of **Earning** with **Point** *is greater than 140* is `430,027`. |
| Numerical Operation (`Diff`) | The difference between `val:1` and `val:2` in `col` is `executed_results`. | The difference between *China* and *Canada* in **Gold** is `16`. |

Table 7: 6 reasoning operations, along with fact template and examples, defined for the fact generation process of REFACTOR. Variable names indicate permissible instantiations. `col` denotes a column name, `val` denotes a cell value, and `executed_results` denotes the execution results of the function. `OPERATOR` is instantiated according to the specific reasoning operation, e.g., for "Numerical Operation", `OPERATOR` is replaced with "sum" or "average"; `CONDITION` can be 1) a cell value from the `i`-th column, or 2) number/temporal comparison statement (e.g. "later than 1967") if the `i`-th column is of number or date type.

```
Table Title: Top 10 Company in 2012

Table:
Rank | Company Name | Industry | Revenue ($ Million) | Profit ($ Million) |
Employees | Headquarters
1 | Walmart | Retail | $559,151 | $13,510 | 2,300,000 | United States
2 | State Grid | Electricity | $386,618 | $5,580 | 896,360 | China
3| Amazon | Retail | $386,064 | $21,331 | 1,298,000 | United States
4 | National Petroleum | Oil and gas | $283,958 |$4,575 | 1,242,245 | China
5 | Sinopec Group | Oil and gas | $283,728 | $6,205 | 553,833 | China
6 | Apple | Electronics | $274,515 | $57,511 | 147,000 | United States
7 | CVS Health | Healthcare | $268,706 | $7,179 | 256,500 | United States
8 | UnitedHealth | Healthcare | $257,141 | $15,403 | 330,000 | United States
9 | Toyota | Automotive | $256,722 | $21,180 | 366,283 | Japan
10 | Volkswagen | Automotive | $253,965 | $10,104 | 662,575 | Germany

Using the information from the table, summarize a paragraph-long
response to the following user query: Which company earns the highest
profit in the Oil and Gas industry, and how does it compare to the most
profitable company overall?
```

Figure 4: An example of LLM zero-shot prompt prefix *wo*. REFACTOR for the QTSUMM task.

```
Table Title: Top 10 Company in 2012

Table:
Rank | Company Name | Industry | Revenue ($ Million) | Profit ($ Million) |
Employees | Headquarters
1 | Walmart | Retail | $559,151 | $13,510 | 2,300,000 | United States
2 | State Grid | Electricity | $386,618 | $5,580 | 896,360 | China
3| Amazon | Retail | $386,064 | $21,331 | 1,298,000 | United States
4 | National Petroleum | Oil and gas | $283,958 |$4,575 | 1,242,245 | China
5 | Sinopec Group | Oil and gas | $283,728 | $6,205 | 553,833 | China
6 | Apple | Electronics | $274,515 | $57,511 | 147,000 | United States
7 | CVS Health | Healthcare | $268,706 | $7,179 | 256,500 | United States
8 | UnitedHealth | Healthcare | $257,141 | $15,403 | 330,000 | United States
9 | Toyota | Automotive | $256,722 | $21,180 | 366,283 | Japan
10 | Volkswagen | Automotive | $253,965 | $10,104 | 662,575 | Germany

Relevant facts of table:
1. The Company Name ordered by Profit ($ Million) is Apple, Amazon,
Toyota, UnitedHealth, Walmart, Volkswagen, CVS Health, Sinopec Group,
State Grid, National Petroleum.
2. The Company Name, with Industry is Oil and gas, ordered by Profit ($
Million) is Sinopec Group, National Petroleum.
3. The sum of Profit with Industry is Oil and gas is 10780.
4. The difference between Apple and Sinopec Group in Profit is 51306.
5. The average of Profit with Industry is Oil and gas is 5390.

Using the information from the table and relevant facts of table, summarize
a paragraph-long response to the following user query: Which company
earns the highest profit in the Oil and Gas industry, and how does it
compare to the most profitable company overall?
```

Figure 5: An example of LLM zero-shot prompt prefix *w*. REFACTOR for the QTSUMM task.

| Query | System output | Reference | Error analysis |
|---|---|---|---|
| *Q*: Which driver performed better, Tracy or Haberfeld, and why is that? | Paul Tracy performed better than Mario Haberfeld, as evidenced in their results from the 2004 Centrix Financial Grand Prix of Denver. In the race, Tracy finished in second place, while Haberfeld placed in eighth. Additionally, Tracy belongs to Forsythe Racing, which has had a more successful track record compared to Haberfeld's Walker Racing. Additionally, Tracy was able to maintain his position in the race for the full 90 laps, whereas Haberfeld was only able to make it to 86 laps. Based on these results, it is clear that Tracy was more competitive compared to Mario Haberfeld. | Paul Tracy is more competitive. Tracy finished the race in second place with a time of +7.446 seconds, whereas Haberfeld finished in eighth place with a time of +24.141 seconds. This demonstrates that Tracy was able to navigate the track much faster than Haberfeld and place higher in the race. Additionally, Tracy earned 28 points while Haberfeld earned only 15 points, which confirms that Tracy is stronger competitor. In conclusion, Tracy is the more competitive driver. | *Error Type*: **Hallucination** 

 *Explanation*: Mention facts that are not shown in the table. 

 *Analysis*: There is no information indicating that Forsythe Racing has a more successful track record compared to Walker Racing, and the model incorrectly states that Haberfeld completed only 86 laps instead of the correct 90. |
| *Q*: According to the voting result, which representative election in North Carolina districts was the most competitive, and why? | The most competitive representative election in North Carolina districts was the one for the 8th district, in which incumbent Larry Kissell (D) lost re-election to Richard Hudson (R). This was the only race which was decided by less than 10 percentage points, with Hudson winning 54.1% of the vote to Kissell's 45.9%. The close margin of the race demonstrates that both candidates were competitive, and it was an intense contest until the very end. | The race in the North Carolina 7th district was the most competitive, as the Democratic incumbent Mike McIntyre won by a slim margin, with only a 0.2% difference between him and his Republican challenger David Rouzer. Furthermore, this election was the only one among all North Carolina districts in 2012 that resulted in a margin of victory within less than 1%. | *Error Type*: **Factual Incorrectness** 

 *Explanation*: Present information that contradicts the source table. 

 *Analysis*: The most competitive representative election should be the North Carolina 7th district, rather than the 8th district, in which the difference between the two rivals was only 0.2 |
| *Q*: Analyse the countries in East Asia with Human Development Index (HDI) higher than 0.8. | The countries in East Asia with Human Development Index (HDI) higher than 0.8 are Hong Kong, Japan, Macau, Mongolia, and South Korea. Hong Kong has the highest HDI at 0.898, followed by Japan at 0.901. Macau has the lowest HDI at 0.653, and North Korea has the highest at 0.897. All of these countries have populations greater than 1.8 million, with the highest population density being in Japan at 337 people per km square. | The countries in East Asia with an HDI higher than 0.8 are Hong Kong (PRC), Japan, and South Korea. Hong Kong has an HDI of 0.898 and a population density per square kilometer of 6390. Japan has an HDI of 0.901 and a population density per square kilometer of 337. South Korea has an HDI of 0.897 and a population density per square kilometer of 500. All three countries have a comparatively high population density which is likely to be a factor in their high HDI. | *Error Type*: **User Intent Misunderstanding** 

 *Explanation*: Include information that is irrelevant to the user question. 

 *Analysis*: The query does not ask for country with lowest HDI, or any country with HDI lower than 0.8. The system output also displays many other errors such as hallucination (confusing South Korea with North Korea) and factual incorrectness (Japan has highest HDI). |
| *Q*: What are the buildings that have been the tallest in Portland, Oregon between 1927 and 1965? Summarize the basic information of these buildings. | The buildings that have been the tallest in Portland, Oregon between 1927 and 1965 are the Public Service Building, the Hilton Portland Hotel, and the Harrison West Condominium Tower. The Public Service Building is located at 920 Sw Sixth Avenue and has been the tallest building in Portland, Oregon. It stands at a height of 220 feet (67 meters) and has 16 floors. The Hilton Portland Hotel is located at 921 Sw Sixth Avenue and has been the tallest building in Portland, Oregon. | Between 1927 and 1965, the tallest buildings in Portland, Oregon were Public Service Building, Hilton Portland Hotel, and Harrison West Condominium Tower. Specifically, Public Service Building was built in 1927, was 220 feet / 67 meters tall, had 16 floors, and held the title of tallest building until 1962. Hilton Portland Hotel was built in 1963, was 241 feet / 73 meters tall, had 22 floors, and held the title of tallest building until 1965. Harrison West Condominium Tower was built in 1965, was 256 feet / 78 meters tall, had 25 floors, and held the title of tallest building until 1969. | *Error Type*: **Repetition** 

 *Explanation*: Generate repetitive information. 

 *Analysis*: The information of these buildings being the tallest in Portland, Oregon has been mentioned repetitively throughout the system output, while the system fail to also distinguish them (until which year each of them was the tallest respectively). |

Table 8: Case study for common errors made by Flan-T5-large *wo*. REFACTOR. The colored text highlights problematic parts of the system output.