# OpenReview forum: "QTSumm: Query-Focused Summarization over Tabular Data"
_EMNLP/2023/Conference — EMNLP 2023 Main_

### Official Review · Reviewer_Pktq · 2023-08-04

**Soundness:** 3

**Excitement:**

2: Mediocre: This paper makes marginal contributions (vs non-contemporaneous work), so I would rather not see it in the conference.

**Paper Topic And Main Contributions:**

This manuscript propose a benchmark for query-focused table summarization.
They also propose table summarization approach named REFACTOR.
Authors find the gap between automatic and human evaluation results and propose QTSUMM-EVAL to investigate the performance of automatic metrics.

**Questions For The Authors:**

Why not compare with other TableQA/Table-to-text generation methods in experiments?
Why only test LLM in the 2-shot setting? Why not use more examples?

**Reasons To Accept:**

This manuscript builds a valuable Table Summarization dataset, which is close to real-world scenarios.
The proposed method achieves the state-of-the-art performance on their dataset.

**Reasons To Reject:**

1) Compared with summarization, it is more like a Table-to-text generation dataset/task. Rather than saying that this article defines a new task, it is more like giving the old task a new name.
2) Compared with previous work on TableQA and Table-to-text generation, the novelty of the proposed method is limited
3) The writing of this manuscript still needs improvements. Many redundant paragraphs can be simplified, and the logic of writing can be clearer.

**Reproducibility:**

4: Could mostly reproduce the results, but there may be some variation because of sample variance or minor variations in their interpretation of the protocol or method.

**Reviewer Confidence:**

5: Positive that my evaluation is correct. I read the paper very carefully and I am very familiar with related work.

---

> ### Author Rebuttal · Authors · 2023-08-29
>
> Thanks for the time you’ve taken to review our work!
>
> ### 1. Concerns about Novelty of QTSumm Benchmark
> > Compared with summarization, it is more like a TableQA/Table-to-text generation dataset/task. Rather than saying that this article defines a new task, it is more like giving the old task a new name. Better to revise the title and submit it to the Question Answering and Natural Language Generation track.
>
> ***Why Submit to Summarization Track***
>
> As discussed in the related work section (line 222-242), query-focused summarization is a crucial sub-field of summarization. The primary objective of this field is to generate summaries from source input that are tailored to user queries. This aligns with the task setting of QTSumm.
>
> ***Comparing QTSumm with Existing Table-Relevant Tasks***
>
> The distinctions between QTSumm and existing Table-to-Text generation / Table QA datasets and tasks have been highlighted in introduction (lines 56-82 and table 1) and the related work (lines 160-200) sections. We copy and organize them as follows, _with corresponding line indices_, for your convenience:
>
> 1. Differences from Table-to-Text Tasks in _Human-like Reasoning and Analysis_:
>    - Existing table-to-text generation tasks (i.e., ToTTo, LogicNLG, and HiTab) focus on converting the highlighted table regions into a _human-readable_ and _coherent single sentence_ (line 60-62, 169-171).
>    - QTSumm emphasized more on human-like reasoning. It focuses on generating _paragraph-long_ and _query-tailored data analysis_ with _human-like reasoning_ process (line 62-68, 171-175).
> 2. Differences from Table Summarization Tasks in _Addressing Information-seeking Purposes_
>    - QTSumm _aligns more closely with real-world scenarios_ of how humans consult tabular data. Considering the numerous data points and diverse user's information needs, it becomes challenging to create a _generic_ summary that encompasses all the salient information within a table (line 181-186).
>    - Existing table summarization tasks (i.e., RotoWire, SciGen, and NumericNLG) use _domain-specific_ tables as source input; while the tables in QTSumm are from Wikipedia and cover _diverse topics_ (line 175-180).
> 3. Difference from Short-form Table QA Tasks in _Addressing More Complex Information Needs_
>    - Existing Table QA tasks (i.e., WikiTableQuestions, WikiSQL, SQA, and HybridQA) focus on retrieving and composing short-form answers (typically one word) that cover a _single_ fact (line 74-77).
>    - QTSumm _aligns more closely with real-world scenarios_, where real-world users often have more complex information needs that expect models to perform _human-like reasoning_ to provide _paragraph-long_ analyses (line 77-82).
> 4. Difference from FeTaQA in _Human-like Reasoning and Analysis_
>    - Most questions in FeTaQA focus on surface-level facts and single-sentence answers (e.g., “Which country held the 2022 FIFA World Cup?”), whereas QTSumm requires models to perform human-like reasoning for implicit data insights (line 196-200).
>    - FeTaQA is collected by rewriting ToTTo’s statements into questions and uses the same statements as the answers; while queries in QTSumm is annotated by human annotators under real-world scenarios, making them more natural and better-reflecting users’ actual information needs (line 188-195).
>
> ***QTSumm Introduces Distinct Challenges in Both Text Generation and Automated Evaluation***
>
> As highlighted in the paper (line 22-25, 144-153), the above unique features of QTSumm introduce new challenges in:
> 1. Text generation (Section 5) - Models must comprehend users' information needs and perform human-like reasoning and analysis over structured tables.
> 2. Automated evaluation (Section 6) - Automated metrics need to reliably assess the faithfulness and comprehensiveness of summaries that are query-tailored and reasoning-required.
>
> &nbsp;
>
> ### 2. Concerns about Experiment Design
> > Why not compare with other TableQA/Table-to-text generation methods in experiments?
>
> We have already evaluated and compared the latest work in TableQA/Table-to-text generation methods (i.e., TAPEX, UnifiedSKG, ReasTAP, PLOG) in section 5 and Table 3. As presented in [1], these models are the state-of-the-art methods in both Table-to-Text generation (i.e., LogicNLG) and Table QA (i.e., WikiTableQuestions, WikiSQL, SQA, and FeTaQA) tasks.
> If the reviewer has additional specific models they’d like to see, we’d be happy to consider adding.
>
> > Why only test LLM in the 2-shot setting?
>
> We have already reported the performance of LLMs in both _0-shot_ and 2-shot settings, with and without the proposed ReFactor method, in section 5 and Table 3.
>
> &nbsp;
>
> ### 3. Concerns about Method Novelty
> > Compared with previous work on TableQA and Table-to-text generation, the novelty of the proposed method is very limited.
>
> **ReFactor is Plug-and-Play, Effective and Adapative** To the best of our knowledge, ReFactor is the first plug-and-play method in table-to-text generation research that can be easily adopted to different types of fine-tuned and large language models (as demonstrated in section 6). Furthermore, in Section 6, we also demonstrate the effectiveness of ReFactor in improving existing table-to-text automated evaluation metrics, making the metric more aligned with human evaluation.
>
> **ReFactor is Proposed for Resolving Common Issues Regarding Implicit Reasoning.** The state-of-the-art TableQA/Table-to-Text methods (i.e., TAPEX, UnifiedSKG, ReasTAP, PLOG) are end-to-end models following the “pre-training and then fine-tuning” paradigm. They generate text in an end-to-end manner, resulting in reduced explainability and difficulties in handling more complex reasoning, such as arithmetic calculation (line 201-221, 379-385, 1212-1226).
> Such "error-prone implicit reasoning" and "reduced explainability" issues motivate us to propose our method, ReFactor, which generates several faithful and query-relevant NL facts serving as _explicit_ reasoning results (line 376-393).
>
> We acknowledge that there are some _task-specific_ methods in Table QA [2-3] that use executable logical forms such as sql query to retrieve and obtain a _single fact_ as the final answer. But we are the first to explore extracting, ranking, and composing multiple facts from tabular data into paragraph-long and query-tailored summary via _explicit_ and _plug-and-play_ reasoning modules. If the reviewer is aware of any specific work that closely resembles ours, we would be happy to review and differentiate our approach accordingly.
>
> &nbsp;
>
> ### 4. Concerns about Writing
> > The writing of this manuscript still needs improvements. Many redundant paragraphs can be simplified, and the logic of writing can be clearer.
>
> Thanks for your suggestions! We will further carefully proofread and improve the manuscript in the revised versions.
>
> &nbsp;
> &nbsp;
>
> ### Reference
> 1. Zhao, Yilun, et al. ["OpenRT: An Open-source Framework for Reasoning Over Tabular Data."](https://aclanthology.org/2023.acl-demo.32.pdf) ACL 2023 demo.
> 2. Herzig, Jonathan, et al. ["TaPas: Weakly supervised table parsing via pre-training."](https://aclanthology.org/2020.acl-main.398.pdf) ACL 2020.
> 3. Nan, Linyong, et al. ["FeTaQA: Free-form table question answering."](https://direct.mit.edu/tacl/article/doi/10.1162/tacl_a_00446/109273/FeTaQA-Free-form-Table-Question-Answering) TACL 2022.

---

### Official Review · Reviewer_vhSV · 2023-08-05

**Soundness:** 4

**Excitement:**

4: Strong: This paper deepens the understanding of some phenomenon or lowers the barriers to an existing research direction.

**Paper Topic And Main Contributions:**

This research introduces a novel task in the field of text generation, focusing on creating tailored summaries for tables based on user queries. The authors propose QTSumm, containing 5,625 query-summary pairs from a diverse set of 2,437 tables. The study explores various strong baselines and evaluation methods, revealing significant challenges in both generating accurate table-to-text summaries and evaluating them. The authors propose a new approach called REFACTOR, which retrieves and reasons over query-relevant information from tabular data to generate natural language facts. Experimental results demonstrate that REFACTOR effectively enhances state-of-the-art models and evaluation metrics by incorporating generated facts into the model input.

**Reasons To Accept:**

Great motivation for building the proposed benchmark: Learning how to summarize structural information with personalized query is very important to let models learn to follow the instructions and provide proper feedback. I’m impressed by Table 3 which contains very comprehensive evaluation and further shows the necessity of having this benchmark.

Comprehensive investigation on the quality of generated instruction data: I'm glad to see that the authors utilize different metrics and many latest models like GPT-4 and Tulu to evaluate the performance. The comprehensive evaluation helps us better understand the effect of the task difficulty and proposed methods.

Good quality control for the benchmark construction: This paper also describes the details of quality control when constructing the dataset. Multi-round validation is helpful for further improving the quality. Also, to make the dataset not that easy, the authors introduce some mechanism to create instances involving multi-hop reasoning.

**Reasons To Reject:**

Lack of details for Refactor in main paper: For Section 4, it is very hard for me to understand the details of Refactor method. Although the authors put description in Appendix B, I may still think generally describing methods with pure text in main paper is not helpful for readers to understand the details. It is better to move them to the main paper, so that readers will know the formulation (e.g., each step’s input/output) more clearly.

**Reproducibility:**

4: Could mostly reproduce the results, but there may be some variation because of sample variance or minor variations in their interpretation of the protocol or method.

**Reviewer Confidence:**

3: Pretty sure, but there's a chance I missed something. Although I have a good feel for this area in general, I did not carefully check the paper's details, e.g., the math, experimental design, or novelty.

---

> ### Author Rebuttal · Authors · 2023-08-29
>
> Thanks for the constructive feedback! We greatly appreciate the time you took to thoroughly read our paper and examine the details in the Appendix. We will relocate "Appendix B.1 - Implementation Details of Fact Generator" to the main body of the paper in the revised version.

---

### Official Review · Reviewer_PL53 · 2023-08-05

**Soundness:** 4

**Excitement:**

4: Strong: This paper deepens the understanding of some phenomenon or lowers the barriers to an existing research direction.

**Paper Topic And Main Contributions:**

The paper presents
(1) a new dataset as well a novel variation of the table-to-text generation problem
by formulating the query-focused summarization of table data where the summaries require human-like reasoning/explanations
rather than facts (different from previous works)
(2) a meta-evaluation benchmark to help in the design of metrics for evaluating the above problem
and (3) a method to summarize table data in response to queries by making use of extracted query-relevant
facts that are ranked
and further combined to generate a summary.

Experiments are presented on this novel dataset with their proposed approach as well as
modifying existing state-of-the-art baselines on closely related problems and the
results quite support the claims made in the paper regarding the importance of
table structure, analytical reasoning, and factual correctness.



**Questions For The Authors:**

Mentioned in previous section

**Reasons To Accept:**

A. The paper is well-written and the proposed problem is a well-motivated extension to
recent research directions on tables-to-text generation as well as query focused summarization

B. The dataset creation process is reliable (multi round validation) and when shared will be a valuable add for further
study on this topic.

C. The method of generating facts relevant to a query and further collating them forms
an explainable way to generate a reliable query focused summary.


**Reasons To Reject:**

A. The paper is somewhat hard to read due to too much going on in the main paper
(several abbreviations, metrics, systems, disparate contributions and referenced details
with having to jump to Appendix to verify some details
that are better fit in the main paper).
For example, just keeping the main dataset and the Refactor method makes a good story
rather than the eval benchmark, and section 6. In particular,
I could not follow much of section 6.2  and what Table-5 with Kendall Tau values
is meant to highlight.

B. For the templates used for fact extraction from tables (Figure 7), there is not much
discussion on what is potential downside.  As in, what types of reasoning
questions may be missed due to reliance on these templates and thoughts on how to overcome them.


**Reproducibility:**

4: Could mostly reproduce the results, but there may be some variation because of sample variance or minor variations in their interpretation of the protocol or method.

**Reviewer Confidence:**

3: Pretty sure, but there's a chance I missed something. Although I have a good feel for this area in general, I did not carefully check the paper's details, e.g., the math, experimental design, or novelty.

---

> ### Author Rebuttal · Authors · 2023-08-29
>
> Thanks for the time you’ve taken to review our work and for the constructive feedback!
>
> ### A. Concerns about Paper Structure
> > The paper is somewhat hard to read due to too much going on in the main paper (several abbreviations, metrics, systems, disparate contributions and referenced details with having to jump to Appendix to verify some details that are better fit in the main paper).
>
> We greatly appreciate the time you took to thoroughly read our paper and examine the details in the Appendix. In the revised version, we will condense and move content from "Appendix A.2 - Details of Automated Evaluation Metrics", "Appendix B.1 - Implementation Details of Fact Generator", and "Appendix C.1 - Details of Baseline Systems" to the main sections of the paper.
>
> > For example, just keeping the main dataset and the Refactor method makes a good story rather than the eval benchmark, and section 6.
>
> Thanks for your suggestion! We will consider it when preparing the final version.
>
> ### B. Questions about Limitations of Proposed Method
> > For the templates used for fact extraction from tables (Figure 7), there is not much discussion on what is potential downside. As in, what types of reasoning questions may be missed due to reliance on these templates and thoughts on how to overcome them.
>
> Thanks for the insightful comments! We appreciate the opportunity to explain and extend this point:
>
> ***B.1 We Add a New Case Study on ReFactor’s Failure Cases***
>
> In section 5.4, we conducted human evaluation on 200 QTSumm examples for assessing the coverage of facts generated by ReFactor. We revealed that 56.4% facts are annotated as “relevant”. To delve deeper into this, we have added a case study examining the failure cases, specifically those examples where two or fewer facts were annotated as “relevant”. The details of this case study are as follows:
>
> | #Examples | Error Types                                                   | Representative Question                                                                                            | Explanation                                                                                                                                                                                                                                                  |
> |--------------|---------------------------------------------------------------|--------------------------------------------------------------------------------------------------------------------|--------------------------------------------------------------------------------------------------------------------------------------------------------------------------------------------------------------------------------------------------------------|
> |       24 / 200      | Difficulty in parsing cell values via rule-based methods      | Difficulty in parsing cell values via rule-based methods                                                           | The relevant numeric- or time-type columns are hard to parse (e.g., multiple numbers and text within one cell), thus ReFactor fail to generate related facts.                                                                                                |
> |      17 / 200      | Complex user query causes difficulty in ranking related facts | Analyze the correlation between the size of the geographical area of a Gmina type and its population?              | ReFactor employs the employs the QA encoding model for fact ranking. However, it struggles to understand complex information needs from users, such as the "correlation between A and B," and might consequently rank irrelevant facts higher. |
> |      13 / 200      | Unsupported reasoning operations                              | Who are the top three coaches with the highest win percentages? Analyze their performance in the 2019-2020 season. | The table only contains “wins” and “overall games” columns. Models must compute the winning percentages independently. However, ReFactor does not support such rate calculations                                                                             |
> |       5 / 200      | Other errors                                                  |                                                                                                                    |                                                                                                                                                                                                                                                              |
> |      141 / 200     | Successful cases                                              |                                                                                                                    |                                                                                                                                                                                                                                                              |
>
> &nbsp;
>
> ***B.2 Thoughts on Future Work***
>
> **The Motivation Behind ReFactor is Compelling.** As discussed in section 4, the motivation of proposing ReFactor is to address the _error-prone implicit_ reasoning issues of current end-to-end text generation models.
> We believe that the high-level ideas of "employing plug-and-play, explicit reasoning modules to extract and reason over query-relevant information as explicit and intermediate reasoning results" is promising for future extensions. Nonetheless, we recognize the inherent limitations of ReFactor, particularly its reliance on predefined and template-based reasoning modules.
>
> **Potential Directions for Future Enhancements.** In the limitation section, we have discussed the potential of applying LLMs with 1) tool learning and 2) multi-step chain-of-thought on generating reasoning-required and query-tailored summary. Given the surge of pertinent research in recent months, we believe the following directions warrant further exploration.
> 1. _Decomposing complex questions into fine-grained sub-questions using chain-of-thought._ Our case study reveals that the TAPEX-based fact ranking module struggles with comprehending complex questions. To address this, future research could investigate LLM chain-of-thought methods to break down complex questions into more understandable and actionable sub-questions.
> 2. _Generating and utilizing executable programs for fact generation._ The predefined and template-based execution modules in the ReFactor fact generation phase have their limitations. Recent studies [1-2] highlight the impressive abilities of LLMs in making and utilizing tools for problem-solving. It would be intriguing to explore if LLMs can produce executable programs from scratch to derive query-relevant insights.
> 3. _Composing insights into paragraph-long analysis using chain-of-thought._ How to use chain-of-thought to compose salient information and generate summary [3] is still underexplored in the community.
>
> We will include and extend above points in the limitation section of the revised version.
>
> &nbsp;
>
> &nbsp;
>
> ### References
> 1. Schick, Timo, et al. ["Toolformer: Language models can teach themselves to use tools."](https://arxiv.org/pdf/2302.04761.pdf) arXiv preprint arXiv:2302.04761 (2023).
> 2. Cai, Tianle, et al. ["Large language models as tool makers."](https://arxiv.org/pdf/2305.17126.pdf) arXiv preprint arXiv:2305.17126 (2023).
> 3. Wang, Yiming, Zhuosheng Zhang, and Rui Wang. ["Element-aware Summarization with Large Language Models: Expert-aligned Evaluation and Chain-of-Thought Method."](https://arxiv.org/pdf/2305.13412.pdf) ACL 2023.

---

### Meta-Review · Area_Chair_jqGU · 2023-09-17

**Recommendation:** 4

**Metareview:**

This paper proposes a new query-focused table summarization task. Unlike other tasks such as Table QA or Table-to-text, the proposed task requires both the adaptation to user queries and the necessary reasoning for summarization.

* Reviewers all agree on the value of the dataset, especially commending the transparency and properness of the construction.
* The authors also presented comprehensive experiments to contrast different SOTA methods for neighbor tasks (i.e., table QA, table summarization).
* At the same time, however, the clarity on the proposed Refactor needs improvement. While some details are presented in the Appendix, the authors are expected to retain the main information in the body of the paper.

---

### Decision · Program_Chairs · 2023-10-07

**Decision:**

Accept-Main

**Comment:**

This paper proposes a new query-focused table summarization task. Unlike other tasks such as Table QA or Table-to-text, the proposed task requires both the adaptation to user queries and the necessary reasoning for summarization.

* Reviewers all agree on the value of the dataset, especially commending the transparency and properness of the construction.
* The authors also presented comprehensive experiments to contrast different SOTA methods for neighbor tasks (i.e., table QA, table summarization).
* At the same time, however, the clarity on the proposed Refactor needs improvement. While some details are presented in the Appendix, the authors are expected to retain the main information in the body of the paper.